# Concentration Distribution and Analysis of Urinary Glyphosate and Its Metabolites in Occupationally Exposed Workers in Eastern China

**DOI:** 10.3390/ijerph17082943

**Published:** 2020-04-24

**Authors:** Feng Zhang, Yanqiong Xu, Xin Liu, Liping Pan, Enmin Ding, Jianrui Dou, Baoli Zhu

**Affiliations:** 1Department of Occupational Disease, Jiangsu Provincial Center for Disease Prevention and Control, Nanjing 210009, China; xyq6562100@sina.com (Y.X.); liuxinjsha@163.com (X.L.); enminding@126.com (E.D.); 2Medical Examination Center, Nanjing Prevention and Treatment Center for Occupational Disease, Nanjing 210038, China; 3Department of Occupational Hygiene, Yangzhou City Center for Disease Prevention and Control, Yangzhou 225000, China; yueer1_18@163.com; 4Center for Global Health, School of Public Health, Nanjing Medical University, Nanjing 210008, China

**Keywords:** workers, glyphosate, aminomethylphosphonic acid, urine, concentration distribution

## Abstract

*Background*: There are few published studies concerning occupational exposure to glyphosate (GLY), and these are limited to spraying, horticulture and other agricultural aspects. Therefore, the concentration of glyphosate and its metabolite aminomethylphosphonic acid (AMPA), in the urine of workers exposed to glyphosate during glyphosate production was determined, and the relationship between internal (urinary glyphosate and AMPA concentration) and external exposure dose (time weighted average (TWA) value of glyphosate in the air of workplace) was analyzed. *Methods*: To avoid the influence of preparations, we selected people who were only involved in GLY production (without exposure to its preparations) as our research subjects. We collected 134 urine samples of workers exposed to GLY (prototype, not preparation). The urinary concentrations of GLY and AMPA (internal exposure dose) were detected by gas chromatography-mass spectrometry. The subjects’ exposure to the amount of GLY in the air (external dose) was determined using ion chromatography. Conventional statistical methods, including quartiles, *t*-tests and regression analysis, were applied for data processing. *Results*: An on-site investigation revealed that the workers involved in centrifugation, crystallization, drying, and packaging and feeding were exposed to GLY. The TWA value of GLY in the workshop air was <0.02 mg/m^3^–34.58 mg/m^3^. The detection rates of GLY and AMPA in the urine samples were 86.6% and 81.3%, respectively. The concentration of urinary GLY was <0.020–17.202 mg/L (median, 0.292 mg/L). The urinary AMPA concentration was <0.010 mg/L–2.730 mg/L (median, 0.068 mg/L). The geometric means were 0.262 mg/L and 0.072 mg/L for GLY and AMPA, respectively. There was a correlation between the urinary concentration of GLY and AMPA and the TWA value of exposed workers (correlation coefficient [r] = 0.914 and 0.683, respectively; *p* < 0.01). Furthermore, there was a correlation between the urinary concentration of GLY and AMPA in the exposure group (r = 0.736, *p* < 0.01). *Conclusions*: The urinary concentration of GLY and AMPA of workers was correlated with the TWA value of workers’ exposure, which could reflect the actual exposure of the workers.

## 1. Background

Since the introduction of glyphosate (GLY) into agricultural practice in the 1970s [1], it has become one of the most widely used herbicides in many countries and regions [2,3]. GLY has been extensively investigated because of its possible adverse health effects in humans, including respiratory system damage, diabetes, Parkinson’s disease, rheumatism, thyroid disease, myocardial infarction and reproductive and developmental system damage [4,5,6]. In recent years, more attention has been paid to the hepatorenal toxicity of GLY [7]. Mesnage et al. [8] revealed that GLY could be toxic if it is below the regular lowest observed adverse effect level for chronic toxic effects, which include teratogenic, tumorigenic and hepatorenal effects. Similarly, Trasande et al. [9] also studied and assessed potential associations with renal function in children. However, there is no evidence in this study for renal injury in children exposed to low levels of GLY, and a larger sample size was needed for further study.

Although more and more cases of health damage from GLY and preparations have been reported in recent years [10,11], there are still many different opinions on its toxicity [12]. The main divergences focus on two aspects. Firstly, the toxicity manifestations and severity of the raw material GLY and commercial GLY preparations are quite different. Some studies thought that the toxicity of GLY came from the surfactant in commercial preparations [13], with the order of toxicity being surfactant > GLY preparation > GLY > GLY isopropylamine salt [14]. However, other scholars deemed that GLY had specific toxicity, and they could not agree whether it was caused by preparations, because the general population rarely comes into contact with pure GLY [15]. Secondly, the results of research on the carcinogenicity of GLY were inconsistent. According to the International Cancer Organization, GLY was classified as probably carcinogenic to humans in 2015 after reviewing existing data [16]. Interestingly, many literature reviews and studies showed that there was no direct evidence of a relationship between GLY and carcinogenesis [17,18]. Despite the controversy over GLY toxicity, the biological monitoring of patients and the general population has continued. 

It was reported that, after oral administration of 14C-labelled GLY preparations in Sprague Dawley rats, 30–36% was absorbed, and <0.27% was scavenged by generating CO_2_. Additionally, aminomethylphosphonic acid (AMPA) was the only metabolite found in urine and feces after repeated administration, and 97.5% of the GLY entering the body was scavenged by maternal compound GLY from the urine and feces [19]. As early as 1991, Jauhiainen et al. [20] reported that the urinary GLY concentration in workers spraying GLY in a forest with a population of 350 trees ranged from 0.01 to 0.1 mg/L, while the GLY concentration in the air ranged from below the detection limit (<1.3 μg/m^3^) to 15.7 μg/m^3^, with most results below the detection limit. Furthermore, a recent article reported that the urinary GLY concentration in children and adolescents was 0.363 ± 0.3210 μg/L and 0.6060 ± 0.5435 μg/L, respectively [21].

There are few published studies on biomonitoring reports of occupational exposure to GLY, and these are limited to spraying, horticulture and other agricultural aspects [20,22,23,24]. In this paper, to avoid the influence of preparations, we selected people who were only involved in GLY production (without exposure to its preparations) as our research subjects. Urine samples were collected at the end of shifts, and the content of GLY and AMPA was determined. The reliability of the internal dose (urinary GLY and AMPA concentration), as an indicator of the exposure dose, was analyzed.

## 2. Materials and Methods

### 2.1. Study Design

#### Selection of GLY Production Facilities

According to the principle of cluster sampling and the results of a previous investigation [25], we selected four enterprises in Jiangsu and Shandong provinces in China that produced GLY. The main principles for site selection were as follows: (1) annual output of GLY >10,000 tonnes; (2) continual production >3 years at a production capacity >80% of the rated capacity; and (3) low mobility of workers and a constant production process.

### 2.2. Subject Selection

A workshop investigation revealed that GLY was formed after a crystallization process. Thus, the workers involved in crystallization, centrifugation (some of the factories used filter), drying and packaging were chosen as subjects. The exclusion criteria were as follows: (1) working life <24 months, (2) exposure to other pesticides during the 2 years prior to the survey and sample collection (exposure frequency >3 times a year and each exposure >8 h), (3) urine specific gravity <1.010 or >1.030, and (4) liver and kidney dysfunction or the administration of hepatotoxic or nephrotoxic drugs within the past 3 months. After the above exclusions, 134 workers met the eligibility criteria and served as the final study cohort. A structured questionnaire was completed by each participant during face-to-face interviews conducted by our topic-based group. The information requested in the questionnaire included demographic characteristics, lifestyle habits (smoking and alcohol consumption), work history, GLY exposure time, exposure to other physical and chemical factors, respiratory protection use, disease history, etc. An informed consent visit was arranged, and the appropriate consent forms were signed to enable the use of the resulting data for research purposes. All participants were rewarded with a free health examination, and the results were kept confidential from others.

### 2.3. Method

#### 2.3.1. Workshop Investigation

On-site consultation and inspection were performed to determine the occupational health status of the chosen GLY production facilities. This included an assessment of the GLY production process: raw material types and consumption; contact modes and GLY exposure, frequency and time; occupational health protection facilities; personal protective equipment; etc.

#### 2.3.2. Determination and Calculation of External Dose

The time-weighted average (TWA) concentration of GLY in the workshop air was determined by personnel monitoring two consecutive shifts so that the external dose of the exposed workers could be calculated as the average value of two shifts per subject. The collector was an ultra-fine glass fiber filter (diameter, 37 mm and aperture, 0.3 μm; Jinnan Company, Nantong, China), and a GilAir-3 sampler (Sensidyne, St. Petersburg, FL, USA) was used [26]. The sampling flow rate was 1 L/min, and sampling continued throughout the entire shift. Modified ion chromatography methods, based on literature [27], were employed to measure the concentration of GLY in the workshop air. The limit of determination (LOD) was 0.02 mg/m^3^, therefore if the test result was lower than the LOD, it was expressed as <0.02 mg/m^3^. The average recovery rates were 95.8%, and the intra-batch and inter-batch precision (relative standard deviation [RSD] %) were 3.7 and 6.2, respectively. At the end of each batch of tests (1 batch = 10 samples), a quality control sample was analyzed. If the standard error exceeded 15%, the samples were retested. We surveyed four GLY production facilities. All of them had obtained a registration license for pesticide production. The production technology was different among each enterprise and divided into iminodiacetic acid and glycine methods. Because GLY was formed after crystallization (filtration), the exposure of workers TWA after the crystallization process was determined. Participants wore their wear work clothes and gloves correctly, but they neglected respiratory protection, and the concentration of GLY in the air inhaled by workers was consistent with the concentration of GLY in the air monitored in the workplace. In addition, the individual measurement method was used to monitor the GLY concentration in the air continuously in the work shift, which can better reflect the actual exposure of workers. Therefore, the TWA values used in the calculation of external doses were without data adjustment and conversion.

#### 2.3.3. Urine Sample Collection and Concentration Determination

The urine samples were collected within 1 h of the end of each of the monitored shifts and measured immediately for specific gravity (results < 1.010 and > 1.030 were excluded). Subsequently, approximately 15 mL of each urine sample was stored at 4 °C for the completion of laboratory testing within 72 h. Otherwise, the samples were stored at −18 °C for the completion of laboratory testing within 14 days. The urinary concentrations of GLY and AMPA were simultaneously determined by derivatization gas chromatography-mass spectrometry, as described previously [28]. The artificial simulated urine sample was used as the blank and was tested before sample analysis. The blank urine samples were lower than the LOD. We used the same derivative method but changed the detector to a mass spectrum to improve sensitivity. Our results showed that the LOD of GLY and AMPA were 0.02 mg/L and 0.01 mg/L, respectively, with the average recovery rates as 94.6% and 92.8%, respectively. The intra- and inter-batch precision (RSD %) of our method was < 8%. A quality control method, similar to that used for air testing, was also performed. When collecting urine samples, subjects were asked to take off their work clothes and wash their hands, arms and faces to avoid GLY contamination on their skin and work clothes.

Because GLY has a certain influence on the kidney function of exposed persons [13,29], a urine specific gravity correction method was used to calculate urinary GLY and AMPA concentration as follows (Equation (1)):(1)$$C_{1}=C_{0}\frac{1.020-1.000}{d-1.000},$$
where *C*_1_ was the GLY or AMPA concentration after correction, *C*_0_ was the measured concentration of GLY or AMPA, and *d* was the specific gravity of the urine samples. The average *C*_1_ value of two shifts also served as the internal dose.

#### 2.3.4. Questionnaire Completion

After the urine sample collection, a structured questionnaire was completed by each participant. The investigators informed the study subjects of the purpose of the study and how their data would be used. All participants expressed their understanding of the information and signed informed consent. The questionnaire included but was not limited to the following aspects: demographic information, type of work, months of exposure to GLY, history of pesticide exposure and use, personal protective equipment, and physical symptoms (self-perception).

### 2.4. Statistical Analysis

IBM^®^ SPSS^®^ Statistics 23.0 (Jiangsu Provincial Center for Disease Control and Prevention network version, AsiaAnalytics, Shanghai, China) and EXCEL 2007 (Microsoft, Redmond, WA, USA) software were used for data analysis. The internal dose was expressed as the median, geometric mean (GM), and quartile. The external dose was expressed as the median and GM. The urinary GLY and AMPA concentrations were compared using non-parametric tests. The Mann–Whitney *U* test with independent samples was applied for comparison between the two groups, and the Kruskal–Wallis test was used for pairwise comparison in multiple groups. Spearman’s correlation test was used to analyze the correlation between the internal and external doses. Continuous data were analyzed by independent sample two-sided *t*-tests if the data were normally distributed and expressed as $\bar{x}\pm \mathrm{standard}\;\mathrm{deviation}$. Otherwise, the quartile representation was used. The α level was set at 0.05. We assigned a value of LOD/2 for concentrations that were below the LOD [30].

## 3. Results

The demographic information of the subjects is shown in Table 1.

The TWA values of subjects exposed to GLY ranged from <0.02 to 34.58 mg/m^3^, and the median was 0.02–12.61 mg/m^3^. The TWA median values of the subjects in packaging positions were the highest, and those in crystallization positions were the lowest (Table 2). The Kruskal–Wallis test results showed that there was a significant difference in TWA among the positions (*H* = 97.009, *p* < 0.01). The results were consistent with the technological process. GLY had just formed in the crystallization section and was in a wet state, so the concentration of GLY in the air was obviously low. Thereafter, the moisture content of GLY decreased gradually during the production process, and the GLY concentration in the powder became higher. Therefore, in the packaging process, the dispersion of GLY was enhanced, and the concentration of aerosols formed in the air was increased. 

Before the formal collection of the urine samples, four workers who participated in the study were asked to provide a minimum of four spot urine samples as follows: (1) before the work task began (pre-task sample), (2) four hours after the start of the work (during shift), (3) within one hour of task completion (post-task sample), and (4) the following morning (following the first morning void). As per Figure 1, the urine GLY and AMPA concentrations collected within one hour after work were clearly the highest. Thus, we decided to collect the formal urine samples during this period to measure GLY and AMPA concentrations. 

We determined the concentration of GLY and AMPA in the urine samples that we collected from the study participants exposed to GLY at different occupational positions (Table 3). The detection rates of GLY (>0.020 mg/L) and AMPA (>0.010 mg/L) were 86.6% (116/134) and 81.3% (109/134), respectively. The median values were 0.292 mg/L and 0.068 mg/L for urinary GLY and AMPA, respectively, and the GM was 0.262 mg/L and 0.072 mg/L, respectively. Consistent with the air concentration distribution, the highest median and GM values of urinary GLY and AMPA were at the packaging post. There was a significant difference in the urinary concentration of GLY and AMPA between the different posts (*p* < 0.05).

Due to the range of urinary GLY and AMPA concentrations over the different posts, we used a logarithmic model to deal with the obtained results to show the characteristics of data distribution (Figure 2). After the logarithmic transformation, we found an outlier, corresponding to the urinary AMPA concentration of 0.418 mg/L. This value is the highest of AMPA concentration among workers in filtration post, but within ± three times of standard deviation of the mean value of this group. Therefore, it was not eliminated in the data statistics.

The internal and external doses of GLY and AMPA in the study cohort of 134 workers are presented in Figure 3. The concentration of glyphosate and AMPA in urine was not a normal distribution, so Spearman’s correlation analysis method was applied. The results showed that there was a correlation between urinary GLY and AMPA concentrations (*r*_s_ = 0.736, *p* < 0.01). We also investigated the correlation between the concentration of GLY in the air and the concentrations of GLY and AMPA in the urine. The concentrations of urinary GLY and AMPA were also positively correlated with TWA. The correlation coefficients were 0.914 and 0.683, respectively, and *p* < 0.01.

## 4. Discussion

Anadón et al. [31] studied the toxicokinetics of GLY and its metabolites in rats and showed that the content of AMPA in plasma was positively correlated with the concentration of GLY after oral administration of a single dose of GLY. The concentration of AMPA and GLY in plasma peaked approximately 4.5 h after administration (100 and 400 mg/kg), and then decreased. The plasma half-lives (beta phase) were 15.08 h and 14.38 h, respectively, but were affected by species, dosage and route of administration. The dosage and route of administration of GLY exert a significant influence. Animal experiments showed that approximately 2% of GLY was bioconverted into AMPA in a single dose (1100 mg/kg per day), and approximately 10–20% of a metabolized GLY prototype was excreted from urine and 80–90% from feces. After stopping administration, the concentration of GLY in urine dropped sharply and comprised mostly of unconverted GLY prototypes [19,32]. The only metabolite that was excreted from the urine within 12 h after exposure was AMPA [19]. 

The urine concentrations of GLY and its metabolite AMPA are often used to assess the degree of GLY exposure. During occupational exposure, GLY enters the human circulation mainly through the respiratory tract, but only ≤2% of the sediment on the skin enters the internal circulation [33]. A review by Gillezeau et al. [34] documented levels of human exposure to GLY among workers in occupational settings and the general population. Urinary GLY concentrations were assessed in 423 people who were occupationally and para-occupationally exposed from eight studied reports, as well as 3298 general subjects’ biological samples from 14 research reports. The average urinary GLY levels in the occupationally exposed subjects ranged from 0.26 to 73.5 μg/L, and the environmental exposure urinary levels ranged from 0.16 to 7.6 μg/L. Only two studies observed the effects of exposure time, both of which showed that the proportion of individuals with detectable levels of GLY in their urine increased with prolonged temporal exposure. Connolly et al. [35] collected 50 urine samples from Irish adults who were non-occupational users of GLY, and only ten contained detectable levels of GLY (range, 0.80–1.35 µg/L). However, the exposure concentrations were higher than those reported in comparable studies of European and American adults. 

Jaime et al. [36] measured the levels of GLY in 81 urine samples from different parts of Mexico. The urine GLY concentration (mean, 0.47 μg/L) in rural areas was higher than in urban areas (mean, 0.22 μg/L). Furthermore, the urinary GLY concentration of certified farmers in California and Minnesota was <1–15 μg/L one day before using a GLY pesticide (one month after the last use and disengagement), while the value on the day of exposure was <1–233) μg/L, and the detectable rate was 60%. However, on the first day after GLY use, the concentration and detection rate of GLY decreased, and on the third day after use, the GLY concentration decreased to <1–68 μg/L, with a detection rate of 27% [30]. On the day of GLY spraying, the detection rate of urinary GLY in the urine of the applicator’s spouse was only 4%, and the highest concentration was 2 μg/L. Nevertheless, one urine sample was found in which the GLY concentration was 1 μg/L during the following two to three days. The results of urinary GLY and AMPA concentration measured in the last three years, as reported in literature, are listed in Table 4. The urinary GLY concentration in our study was higher than that reported in literature. The main reasons were as follows: (1) The subjects of this study were had all been exposed to GLY for at least one year continuously, which accumulated in the body after repeated exposure; (2) The concentration of GLY in the air was relatively high; (3) The subjects did not have personal respiratory protective equipment and inhaled glyphosate directly; (4) Most of the GLY contact in literature came from 10% preparations that were used for spraying. The GLY raw materials that our study subjects came into contact with existed in the air in the form of dust. Compared with the water solution sprayed as pesticides, it was easier to enter for the dust the human body through the respiratory tract.

Concentrations of GLY and AMPA in blood and urine are commonly applied as clinical indicators of the degree of GLY poisoning. Zouaoui et al. [40] detected GLY and AMPA in urine and blood of patients with GLY poisoning and found that the severity of poisoning was positively correlated with the concentration of GLY and AMPA in urine and blood. The concentration of GLY in the urine of patients with mild poisoning did not exceed 3000 mg/L, but it reached 15,000 mg/L in the urine of patients with severe poisoning. The urinary concentration of GLY decreased rapidly with a delay in the sampling time. In three patients with GLY poisoning detected by nuclear magnetic resonance spectroscopy, the concentration of GLY in urine samples was 5746–211,125 mg/L, which was much higher than the highest value obtained in our study (17.202 mg/L). Therefore, our research subjects did not show any symptoms associated with GLY poisoning [41]. However, some subjects also showed the health damage caused by long-term exposure to GLY, which will be the focus of our next study.

Conrad et al. [39] studied the correlation between 24-h urinary GLY concentration and AMPA concentration in a general population of 399 Germans (non-vegetarians). They found a positive correlation between GLY and AMPA concentration (*r* = 0.506, *p* < 0.001), which was consistent with the results of our study. The ratio of glyphosate to AMPA concentration in urine was higher (median was 3.9) compared with literature [39]. The higher ratio in this paper may be related to the collection time and the original GLY concentration. In this study, the higher concentration of GLY exposure and the longer metabolizing time may be needed. Some GLY may not be metabolized in urine samples collected one hour after exposure, resulting in a higher ratio of GLY and its metabolite concentration than that of urine samples collected 24 h later. Cho et al. [42] measured the urinary concentration of GLY and AMPA in a patient with acute GLY potassium poisoning and found that the concentration of GLY decreased rapidly after oral administration of GLY for six hours [42]. According to the measured data, the concentration of GLY was one order of magnitude higher than that of AMPA, which was similar to our study results.

## 5. Conclusions

This study assessed the relationship between the concentration of GLY in the air and the concentration of GLY and its metabolites in biological samples. The results suggested that the urinary concentration of GLY and AMPA was positively correlated with the concentration of GLY in the air, and the median concentration of urinary GLY and AMPA was consistent with the median concentration of GLY in the air, which could reflect the actual exposure of workers. In our future research, we will study occupational exposure limits of GLY from the relationships of health effects and internal and external dose.

## Figures and Tables

**Figure 1 ijerph-17-02943-f001:**
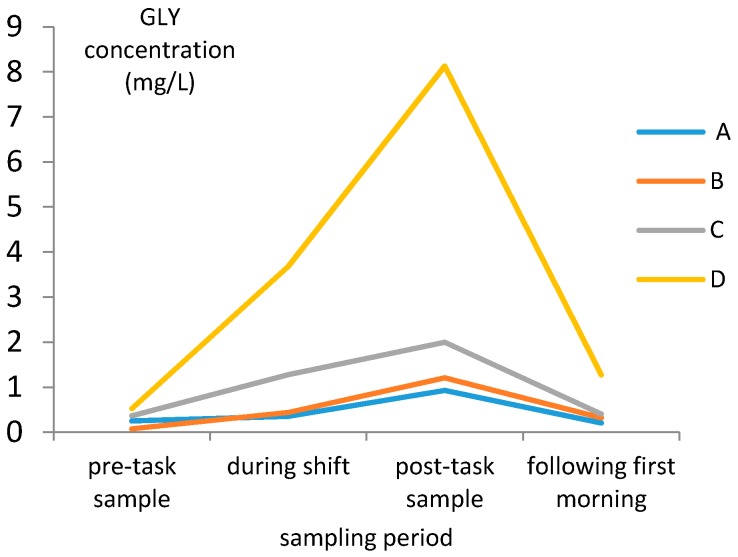
Concentration of GLY and aminomethylphosphonic acid (AMPA) in urine collected in different time periods. (A, B, C and D represents four different sampling objects).

**Figure 2 ijerph-17-02943-f002:**
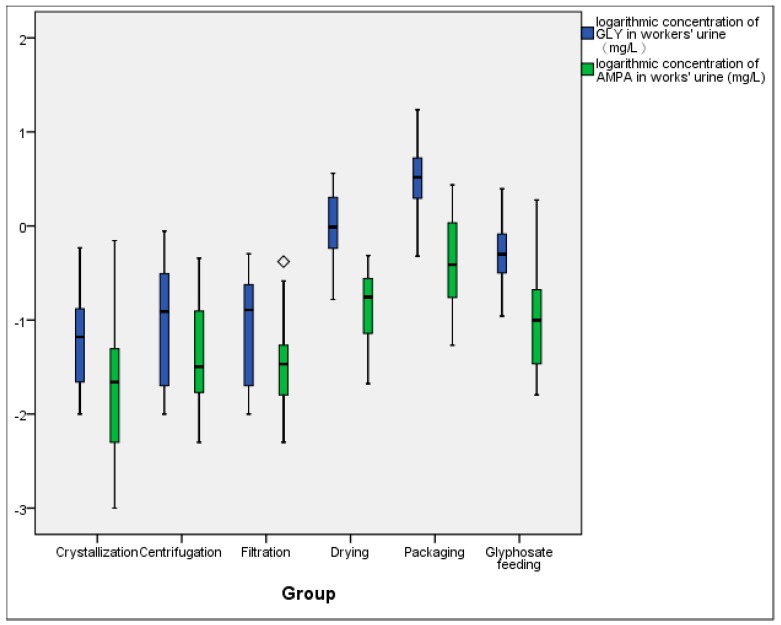
Distribution of urinary glyphosate and AMPA according to the different occupational groups.

**Figure 3 ijerph-17-02943-f003:**
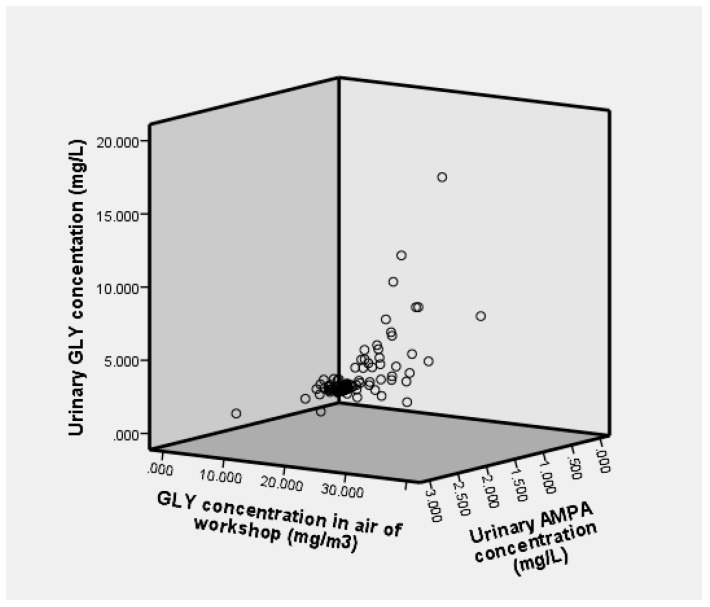
Three-dimensional distribution of internal and external dose.

**Table 1 ijerph-17-02943-t001:** The information of the subjects (N = 134).

Features	Information
Age (years)	41.1 ± 8.1
Length of employment (years)	16.1 ± 10.4
Duration of GLY exposure	8.2 ± 3.7
Smoking [n (%)]	57 (42.5)
Drinking [n (%)]	70 (52.2)
Male [n (%)]	112 (83.6)
Female [n (%)]	22 (16.4)

GLY, glyphosate.

**Table 2 ijerph-17-02943-t002:** Time-weighted average (TWA) values of subjects exposed to glyphosate (GLY) per position.

Position	Population Size	Concentration Range (mg/m^3^)	Median (mg/m^3^)	GM (mg/m^3^)
Crystallization	28	<0.02–0.33	0.02	0.04
Centrifugation	32	<0.02–2.15	0.13	0.11
Filtration	17	<0.02–0.28	0.07	0.07
Drying	22	0.63–8.73	3.72	2.58
Packaging	27	1.28–34.58	12.61	11.78
Raw material feeding	8	0.23–6.95	1.49	1.32

GM, geometric mean.

**Table 3 ijerph-17-02943-t003:** Concentrations and descriptive statistics for urinary GLY and AMPA in urine.

Positions	No.	GLY (mg/L)						AMPA (mg/L)					
		Range of Concentration	Percentile				GM	Concentration Range	Percentile				GM
			*P* _25_	*P* _50_	*P* _75_	*P* _95_			*P* _25_	*P* _50_	*P* _75_	*P* _95_	
Total	134	<0.020–17.202	0.063	0.292	1.254	5.127	0.262	<0.010–2.730	0.023	0.068	0.259	1.161	0.072
Crystallization	28	<0.020–0.586	0.023	0.067	0.129	0.428	0.061	<0.010–0.699	0.005	0.022	0.045	0.444	0.024
Centrifugation	32	<0.020–0.881	0.020	0.123	0.308	0.781	0.084	<0.010–0.454	0.018	0.032	0.116	0.273	0.038
Filtration	17	<0.020–0.505	0.020	0.128	0.238	0.371	0.076	<0.010–0.418	0.016	0.034	0.054	0.292	0.036
Drying	22	0.165–3.629^▲bcd^	0.596	0.974	1.879	2.834	0.914	0.021–0.485^▲bcd^	0.072	0.176	0.273	0.354	0.117
Packaging	27	0.478–17.202^▲bcde^	1.967	3.297	5.342	10.546	2.997	0.054–2.730^▲bcde^	0.174	0.388	1.108	2.669	0.419
Glyphosate feeding	8	0.110–2.491^▲bcdef^	0.364	0.501	0.722	2.034	0.511	0.016–1.891^▲bf^	0.031	0.111	0.245	1.891	0.106
*H* ^a^		89.234						52.976					
*P* ^a^		<0.01						<0.01					

GLY, glyphosate; AMPA, aminomethylphosphonic acid; GM, geometric mean. ^a^ Results of Kruskal–Wallis test of positions in different groups. ^▲^ Mann–Whitney U test. ^b^ Compared with crystallization position *P*^b^ < 0.05. ^c^ Compared with centrifugation position *P*^c^ < 0.05. ^d^ Compared with filtration position *P*^d^ < 0.05. ^e^ Compared with drying position *P*^e^ < 0.05. ^f^ Compared with packaging position *P*^f^ < 0.05.

**Table 4 ijerph-17-02943-t004:** The urinary GLY and AMPA concentration in some recent research studies.

Publication	Population	Population Size	GLY Mean (Range)	AMPA Mean (Range)
Jaime et al., 2017 [36]	Subsistence farmers	81	Median 0.2585 (<0.05–0.7959) μg/L	–
Knudsen et al., 2017 [37]	Children	13	1.28 (0.49–3.22) μg/L	–
	Mothers	14	1.96 (0.85–3.31) μg/L	–
Parvez et al., 2018 [38]	Pregnant women	71	3.40 (0.5–7.20) μg/L	–
Connolly et al., 2017 [22]	Horticultural workers using sprayers	17	0.71 (0.13–3.43) μg/L prior to spraying 1.35 (0.12–10.66) μg/L after spraying	–
Conrad et al., 2017 [39]	German adults	40	Median <0.1 (<0.1–0.57 μg/L), 2015	Median <0.1, (<0.1–0.41) μg/L, 2015
Sierra-Diaz et al., 2019 [21]	Children and adolescents residing in two agricultural communities	140	0.363 μg/L	
		89	0.606 μg/L	

## Data Availability

The datasets used and/or analyzed during the current study are available from the corresponding author on reasonable request.

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
