# Peer review of "Concentration Distribution and Analysis of Urinary Glyphosate and Its Metabolites in Occupationally Exposed Workers in Eastern China"

_ijerph, 2020, doi:10.3390/ijerph17082943_

Round 1
Reviewer 1 Report
The study by Feng et al. looks at concentrations of glyphosate and AMPA in the urine of workers exposed to glyphosate manufacturing and concentrations in ambient air. While the authors do not clearly state the aim of the paper, I assume it is to investigate glyphosate risks and exposure during the production process. I am not quite sure of the purpose of not investigating/avoiding the influence of glyphosate preparations. Is this because from a the toxicity point of view, results may be influenced by the use of surfactants during preparation? I do not see this an issue with direct chemical analysis. The exposure route during preparation (which would be the exposure route most humans would likely have from using the product) is a highly important one too. Nevertheless, I recognise the importance and relevance of this study and its findings. The study seems sound for the most part and I see merit in its publication.
Some concerns I have include: the concentrations reported in this study and some in the literature values (i.e. in ppm (mg/L)) based on literature I have looked at, seem extremely high and I would suggest the authors check this very carefully. The authors report concentrations from a study in Mexico (i.e. Erick Sierra-Diaz; reference #15) in line #76 : “Mexico was 0.363 ± 0.3210 mg/L and 0.6060 ± 0.5435 mg/L, respectively [15].” However, when this study actually reported concentrations in ng mL (ppb). In Table 4 you report these levels again in ug/mL levels. This is a huge error that makes me not trust the data anymore. I would suggest the authors go back and investigate the units of all the literature values they report here as well as their own data accordingly. I do not see how this paper can be published before this is clarified.
There is no mention of blank / control samples as part of QAQC. Can you please clarify how these were incorporated.
Please visit all numbers reported and only report significant figures.
Line 21 – “Therefore, measured”… who measured? Please check through the grammar and English throughout the manuscript for these types of errors.
Line 33 – what does “input” mean?
Lines 34-44 – please watch and correct significant figures through the manuscript (i.e. 17.202 and 0.779)
I am curious how the ratio of Gly – AMPA is so low. Urinary concentration of 17.2 mg/L are extremely high. Would it be worth looking at urine several hours after exposure (or every 1-2 hours after exposure for 6-8 hours?
Line 38 – 0072? I suggest you proof read the manuscript carefully for these mistakes
Lines 138-140 – how were these recoveries analysed? Did you spike before derivatisation? How do you account for the efficiency if the derivatization process.
Figure 2 (B) – correlation between Gly and AMPA. There does not appear to be a good correlation in this figure at all (r2=0.27). Can you investigate what is happening when you only look up until 2.5 on x axis, similarly what happens when you look up until 5 on the x axis? The few rogue numbers at the top of the graph seem to be giving false regression, and even with these values an r2 = 0.27 does not indicate a good correlation.
Author Response
Thank you for your comments on this manuscript. The main purpose of this paper was to explore the concentration distribution of glyphosate and its metabolite AMPA in the population exposed to glyphosate, and the ultimate purpose was to assess the health risk of the population exposed to glyphosate (this part has been described in other studies). As you pointed out, due to the influence of surfactants, there are many studies on the toxicity of glyphosate preparation and many debates. Therefore, in order to avoid the interference of the toxicity of surfactants, this paper chooses the people who are exposed to glyphosate rather than its preparation as the research object.
- About the units and values in this paper
Reply: We downloaded the latest version of the document and found that the concentration unit of glyphosate really changed to ng/mL, which may be related to the online prerelease version we referred to when writing this article. We revised the mg/L (i.e., μg/mL in Table 4) the original manuscript to μg/L. Two versions of the literature are attached.
As for why the research results of this paper are higher than those of other literatures, it is involved in the discussion part. As following:
The urinary GLY concentration in our study was higher than that reported in the literature. The main reasons were as follows: (1) The subjects of this study were all people who have been exposed to glyphosate for at least one year continuously and accumulated in the body after repeated exposure; (2) The concentration of glyphosate in the air was relatively high; (3) The subjects did not have personal respiratory protective equipment and inhaled glyphosate directly; (4) Most of the GLY contact in the literature came from 10% preparations that were used for spraying. The GLY raw materials that our study subjects came into contact with existed in the air in the form of dust. Compared with the water solution sprayed as pesticides, it was easier to enter the human body through the respiratory tract.
- There is no mention of blank / control samples as part of QAQC. Can you please clarify how these were incorporated.
Reply: In our study, 152 non-contact glyphosate workers were selected as control group. The concentration of GLY in urine was <0.020-0.348mg/L, and AMPA was <0.010-0.187mg/L. The 152 people were mainly from the sales and administrative personnel of the factory, not included in the 134 people in the manuscript. It was considering that this paper was to study the concentration distribution of the exposed population, so this control data were not included in the article. However the data of the control population may be more used for health risk assessment.
- Visit all numbers reported and only report significant figures.
Reply: Significant figures were listed in bold in the revision.
- Line 21 – “Therefore, measured”… who measured? Please check through the grammar and English throughout the manuscript for these types of errors.
Reply: We prefixed the sentence with a subject and changed it to “Therefore,the concentration of glyphosate and its metabolite aminomethylphosphonic acid (AMPA), in urine of workers exposed to glyphosate during glyphosate production was determined, and the relationship between internal (urinary glyphosate and AMPA concentration) and external exposure dose (time weighted average [TWA] value of glyphosate in the air of workplace) was analyzed.”
- Line 33 – what does “input” mean?
Reply: Input means adding glyphosate to the reactor, and we changed it to “feeding”
- Lines 34-44 – please watch and correct significant figures through the manuscript (i.e. 17.202 and 0.779)
Reply: Because the concentration of this study is several orders of magnitude higher than that reported in the previous literature. The results of this study are expressed in two concentration units, that is, mg/L is used for the results of this study, and μg/L is used uniformly in the literature. 1mg/L = 1000μg/L.
- I am curious how the ratio of Gly – AMPA is so low. Urinary concentration of 17.2 mg/L are extremely high. Would it be worth looking at urine several hours after exposure (or every 1-2 hours after exposure for 6-8 hours?
Reply: This opinion is very good, which is very valuable for the later research on the metabolism and excretion process of glyphosate in the body. However, I don't quite understand what you mean by the low GLY-AMPA ratio. In this study, according to the literature reports, samples within 1 hour after exposure were collected. Because the samples have been collected, the data of other time periods have not been obtained. The ratio of glyphosate to AMPA concentration is very high compared with the literature. The higher ratio in this paper may be related to the collection time and the original glyphosate concentration. In this study, the higher concentration of glyphosate exposure and the longer metabolizing time may be needed. Some glyphosate may not be metabolized in urine samples collected one hour after exposure, resulting in a higher ratio of glyphosate and its metabolite concentration than that of urine samples collected 24 hours later. This part is supplemented in the discussion. Moreover, GLY is mostly excreted in prototype, and the proportion of metabolism to AMPA is very low.
- Line 38 – 0072? I suggest you proof read the manuscript carefully for these mistakes
Reply: This should be 0.072 mg/L. We have rechecked all the data in the article and unified the units in the article to the same level.
9 Lines 138-140 – how were these recoveries analysed? Did you spike before derivatisation? How do you account for the efficiency if the derivatization process.
Reply: We spiked before derivatisation, and the recoveries steps are as follows:
We applied a similar derivation process in the literature. But in the process of derivatization, we consider the influence of temperature and time. The results showed that when the temperature of derivatization increased, the response value increased, but when it was higher than 100 ℃, the efficiency of derivatization was only 2% higher than that at 90 ℃, so the temperature of derivatization was 90 ℃. We also choose 60℃ as the derivative time.
- Figure 2 (B) – correlation between Gly and AMPA. There does not appear to be a good correlation in this figure at all (r2=0.27). Can you investigate what is happening when you only look up until 2.5 on x axis, similarly what happens when you look up until 5 on the x axis? The few rogue numbers at the top of the graph seem to be giving false regression, and even with these values an r2 = 0.27 does not indicate a good correlation.
Reply: We changed the statistical method. The concentration of glyphosate and AMPA in urine was not normal distribution, so spearmen correlation analysis method was applied. The results showed that there was a certain correlation between the concentration of glyphosate and AMPA in urine (rs=0.736, P<0.01). We also investigated the correlation between the concentration of glyphosate in the air and the concentration of glyphosate and AMPA in the urine. The results also showed that there was a certain correlation, and the correlation coefficients were 0.914 and 0.683, respectively. The statistical values were both P<0.01. We changed Figure 2 to a three-dimensional distribution of internal and external dose (Figure 3 in the revised version).
Reviewer 2 Report
- Page 2, line 62-63: the risk ratio or odds ratio values in many studies was contained 1 (95% confidence interval)… I suggest authors should clarify the statement.
- Did you actually apply all these Statistical analysis tools in your study?
- Page 4, line 173-178: These statement should be under methodology
- Where is the summary of The Kruskal–Wallis test results to show the significant difference in TWA among the positions?
- Figure 1 is wrongly interpreted. Box plot tells you what your outliers are and their values. The authors need to interpret it with 95% CI that the true value of mean are differ or not.
- I suggest authors should validate their regression analysis results to better explain the relationship by using coefficient of determination or any other.
Author Response
Thank you for your valuable comments on this paper. We have replied one by one.
- Page 2, line 62-63: the risk ratio or odds ratio values in many studies was contained 1 (95% confidence interval)… I suggest authors should clarify the statement.
Reply: We change a description method to directly explain the research conclusion of the literature, and delete “the risk ratio or odds ratio values in many studies was contained 1 (95% confidence interval)”
- Did you actually apply all these Statistical analysis tools in your study?
Reply: After verification, we have applied the methods listed in statistical analysis, but changed Pearson correlation to Spearman correlation analysis, because the data is not normal distribution
- Page 4, line 173-178: These statement should be under methodology
Reply: We have placed this part under external dose measurement and calculation of method
- Where is the summary of The Kruskal–Wallis test results to show the significant difference in TWA among the positions?
Reply: We added the result of K-W test: The Kruskal – Wallis test results showed that there was a significant difference between different groups, H=97.009, p<0.01.
- Figure 1 is wrongly interpreted. Box plot tells you what your outliers are and their values. The authors need to interpret it with 95% CI that the true value of mean are differ or not. I suggest authors should validate their regression analysis results to better explain the relationship by using coefficient of determination or any other.
Reply: We supplement the description of outliers and use the nonparametric test Spearman to reanalyze their correlation, and reinterpret the results. As following:
Due to the range of urinary GLY and AMPA concentrations over the different posts, we used a logarithmic model to deal with the obtained results to show the characteristics of data distribution (Fig. 2). After the logarithmic transformation, we found an outlier, corresponding to the urinary AMPA concentration of 0.418mg/L. This value is the highest of AMPA concentration among workers in filtration post, but within ± 3 times of standard deviation of the mean value of this group. Therefore, it was not eliminated in the data statistics.
Reviewer 3 Report
Please see attached the Review Report file for the comments and suggestions.

Round 2
Reviewer 1 Report
The Authors have for the most part adequately taken on the suggestions of all the reviewers and have improved the quality of the manuscript.
However, the authors still do not provide data on important analytical QAQC procedures. Specifically:
1) No data is given on analytical procedural blanks (i.e. if there were detected Gly and AMPA in analytical BLANKS; and if so what were these levels).
2) Similarly what were the %recoveries for Gly and AMPA labelled surrogates following derivetisation and analysis. The Authors indicated some of this in their response (i.e. some %recoveries were 2%), but not in the manuscript.
These data need to be included in the manuscript as they from the basis of the reported results and they are quite important for understanding the quality of the data that is reported. I would strongly suggest therefore that they are included prior to publication.
Author Response
Point 1: No data is given on analytical procedural blanks (i.e. if there were detected Gly and AMPA in analytical BLANKS; and if so what were these levels).
Response 1: The artificial simulated urine sample is used as the sample blank and is tested before sample analysis. The blank urine samples were lower than the detection limit.
Point 2: Similarly what were the %recoveries for Gly and AMPA labelled surrogates following derivetisation and analysis. The Authors indicated some of this in their response (i.e. some %recoveries were 2%), but not in the manuscript.
Response 1: Your question is very good. We only studied the recovery rate of glyphosate and AMPA prototype in the experiment, with an average recovery rate of 94.6% and 92.8%. This part has been included in the article. However, there is no study on the recovery of the internal standard labelled surrogates. The literature of the same internal standard was consulted (Listed in attachment), and the recovery rate of the internal standard was not studied, but the better recovery rate was also obtained. But this reminds us of the way of thinking in the future research. But this reminds us of the way of thinking in the future research.
The 2% you point out is not a recovery, but a derivative temperature effect. when temperature of derivatization was higher than 100 ℃, the efficiency of derivatization was only 2% higher than that at 90 ℃
